# Characterization of the Community of Black Meristematic Fungi Inhabiting the External White Marble of the Florence Cathedral

**DOI:** 10.3390/jof9060665

**Published:** 2023-06-13

**Authors:** Letizia Berti, Massimiliano Marvasi, Brunella Perito

**Affiliations:** 1Department of Biology, University of Florence, Via Madonna del Piano 6, Sesto Fiorentino, 50019 Florence, Italy; letizia.berti@uniroma1.it (L.B.); massimiliano.marvasi@unifi.it (M.M.); 2Department of Sciences of Antiquity, “La Sapienza” University of Rome, Piazzale Aldo Moro 5, 00186 Rome, Italy

**Keywords:** marble biodeterioration, meristematic black fungi, phylogenetic analysis, essential oils, biocide sensitivity, stone cultural heritage conservation

## Abstract

Meristematic black fungi are a highly damaging group of microorganisms responsible for the deterioration of outdoor exposed monuments. Their resilience to various stresses poses significant challenges for removal efforts. This study focuses on the community of meristematic fungi inhabiting the external white marble of the Cathedral of Santa Maria del Fiore, where they contribute to its darkening. Twenty-four strains were isolated from two differently exposed sites of the Cathedral, and their characterization was conducted. Phylogenetic analysis using ITS and LSU rDNA regions revealed a wide diversity of rock-inhabiting fungal strains within the sampled areas. Eight strains, belonging to different genera, were also tested for thermal preferences, salt tolerance, and acid production to investigate their tolerance to environmental stressors and their interaction with stone. All tested strains were able to grow in the range of 5–30 °C, in the presence 5% NaCl, and seven out of eight strains were positive for the production of acid. Their sensitivities to essential oils of thyme and oregano and to the commercial biocide Biotin T were also tested. The essential oils were found to be the most effective against black fungi growth, indicating the possibility of developing a treatment with a low environmental impact.

## 1. Introduction

Biodeterioration of stone monuments exposed in outdoor urban environments has increased over the years due to increased bioreceptivity of materials boosted by atmospheric pollutants [1,2].

Among the stone biodeteriogens, black meristematic fungi occupy a central role. They are widely spread on natural and cultural stone materials [3]. They have evolved and adapted to live on rock surfaces and in very harsh conditions, such as extreme temperatures, drought, starvation, osmotic stress, and UV radiation [4,5,6,7,8]. Black meristematic fungi are also known as rock-inhabiting fungi (RIF) [9,10,11]. RIF are characterized by thick melanized cellular walls and a very slow growth rate [4] and meristematic development [12]. These elements can explain their resistance. For example, melanins are a determining factor as responses to chemical and physical stress [13,14,15].

In addition to aesthetic chromatic alterations due to the presence of melanins (i.e., darkening) [16], meristematic fungi are responsible of mechanical stress that cause deterioration [17,18]. When colonizing rocks, RIF can penetrate into the porosity and cracks, causing micropitting [17,18]. Furthermore, their growth among crystals on the rock surface can cause stone pulverization [19]. Recently, some meristematic strains were found positive for acid production, with potential for a chemical damage to the lithic surface; moreover, their deteriogenic potential could be extended to other materials, since they were found positive for esterase, amylase, cellulase, caseinase, and pectinase activities [20,21]. Therefore, RIF are considered as one of the most damaging groups of microorganisms, causing deterioration of outdoor exposed monuments [16,22,23]. The presence of these microorganisms poses a challenge for restorers, as they are difficult to remove from cultural heritage materials. Moreover, their resistance to various chemical treatments further complicates the restoration process [24].

Even if RIF appear morphologically similar, they are phylogenetically heterogeneous. Therefore, a phylogenetic analysis is often necessary to estimate their biodiversity [4,25]. RIF are classified in the phylum of Ascomycota, in the two classes of Eurotiomycetes and Dothideomycetes. The most common orders are Capnodiales, Chaetothyriales, and Mycosphaerellales [26,27,28]. Up to now, more than 175 species have been identified [27].

Many meristematic fungi were isolated from marble monuments from different heritage sites of the Mediterranean basin, where RIF spread is favoured also by temperature and exposure conditions, such as in Greece [17,29], in Turkey [30], in the Vatican City, and in Italy [6,19,23].

In previous works, we analyzed the microbial community associated with the darkening of the external white marble of the Cathedral of Santa Maria del Fiore (SMFC) in Florence (Italy) through cultivation [2] and targeted metagenomics [31]. We demonstrated that black fungi, together with dark cyanobacteria, were the microorganisms mainly responsible for the darkening of marble [2]. The fungal community inhabiting the darkened areas of marble was first investigated by cultivation conditions, favouring the fast-growing filamentous fungi; in these conditions, only a few black yeasts and meristematic fungi were isolated [2]. On the other hand, the metataxonomic analysis of the microbial community revealed the meristematic fungi as the main components of the fungal community [31].

To date, very little information is available on RIF sensitivity to biocides. A few studies have been performed, testing traditional biocides [20,21,24]. Commercial biocides commonly used in conservation are very often toxic for human health and the environment and might be aggressive for materials, too. Moreover, they are nonspecific, so they could increase bioreceptivity [32]. For these reasons, innovative green methods have been tested for biodeterioration control. Among them, essentials oils (EOs) have been extensively considered as biocides of natural source against biodeterioration of stone cultural heritage due to their well recognized antimicrobial activity, which was investigated in several fields [33]. The inhibition effect of EOs have been tested against single strains (e.g., [34,35]) and the whole cultivable community [36] of filamentous fungi and bacteria inhabiting cultural heritage stone. There is no information, however, about the sensitivity of meristematic fungi towards EOs, despite their role in stone biodeterioration.

This work reports a more in-depth investigation of the community of meristematic fungi inhabiting the external white marble affected by darkening in two differently exposed sampling sites of the Florence Cathedral. The RIF were isolated from marble by suitable cultivation conditions, and a total of 24 strains were analyzed by a multilocus phylogeny. Among these, eight phylogenetic representative strains were further characterized for their physiological properties and their sensitivity to EOs.

The data obtained are a contribution to the knowledge on the biodiversity of meristematic fungi dwelling on heritage marble in the urban environment and to planning innovative low-impact strategies to control their growth and to preserve the marble of Florence Cathedral and other similar structures.

## 2. Materials and Methods

### 2.1. Sampling and Cultivation of Meristematic Fungi

The two study sites, one exposed to the northwest (NW) and the other exposed to the southeast (SE), are part of the external gallery, running around the apses on the upper part of SMFC, as already described [2,37]. Sampling was performed from marble surface areas of the inside face of the openwork parapet of the gallery affected by darkening (Figure 1; [37]) and was described in detail by Checcucci et al. (2022) [31]. At each study site, three samples of about a few tens of mg each of superficial particulate were gently scraped from marble with a sterile spatula (micro-invasive method) in three random points from a surface of about 1000 cm^2^ and collected into sterile tubes. The three samples were pooled in equal proportions in the laboratory and used as a single sample, representative of the whole surface of 1000 cm^2^. This sample was processed for cultivation of filamentous fungi, as described in Santo et al. (2021) [2], as well as for total DNA extraction for metataxonomic analysis, as described by Checcucci et al. (2022) [31]. An aliquot of the same sample was used for cultivation of meristematic fungi (present work), as follows: 15 mg were suspended in 1.5 mL of Phoshate Buffered Saline (PBS; 8 g/L NaCl, 0.2 g/L KCl, 1.44 g/L Na_2_HPO_4_, g/L 0.24 KH_2_PO_4_, pH 7.4), with 0.001% (*v*/*v*) Tween 80 added and vortexed. An amount of 0.1 mL of undiluted and 10^−1^ diluted suspension in PBS were plated in triplicate on Dichloran Rose Bengal Chloramphenicol (ISO) Agar Base (DRBC, Oxoid) and then incubated at room temperature (RT) up to eight weeks. The viable titer was calculated as mean value of the number of Colony Formant Units (CFUs) per gram of marble particulate.

As soon as black colonies were visible, they were transferred onto fresh MEA no. 2 (VWR Chemicals, Milan, Italy). Morphology of colonies grown on MEA after 28 days of incubation was observed under the stereomicroscope Olympus SZX9 (Olympus, Tokyo, Japan). To classify colonies, color, surface morphology, mycelium development at the center and periphery, and size were observed. Colonies exhibiting similar characteristics were grouped together into distinct morphotypes. For each morphotype, at least one strain was re-isolated on DRBC and identified by rDNA analysis.

### 2.2. DNA Extraction and PCR Amplification

DNA was extracted from a mycelium grown on DRBC plates using the Wizard^®^ Genomic DNA Purification kit (Promega, Madison, WI, USA), following the yeast protocol modified in the cell lysis step, as described by Santo et al. (2021) [2]. PCR reactions were performed by using the VWR Taq DNA Polymerase Master Mix (VWR International Srl, Milan, Italy), according to the manufacturer’s instructions, including the use of 1 or more μL of extracted DNA and 0.4 µmol of each primer. Amplification of the internal transcribed spacer (ITS) region of rRNA genes was performed using the forward primers EF3RCNL (5′-CAAACTTGGTCATTTAGAGGA-3′) or ITS1D (5′-GTTTCCGTAGGTGAACCTGC-3′) and the reverse primer ITS4 (5′-TCCTCCGCTTATTGATATGC-3′). When the primers EF3RCNL and ITS4 were used, PCR conditions consisted of an initial denaturation step at 94 °C for 2 min, followed by 35 cycles at 94 °C for 1 min, 53 °C for 1 min, 72 °C for 2 min, and a final extension at 72 °C for 10 min. When using the primers ITS1D and ITS4, PCR conditions consisted of an initial denaturation step at 95 °C for 2 min, followed by 35 cycles at 95 °C for 30 s, 60 °C for 1 min, 72 °C for 1 min, and a final extension at 72 °C for 5 min. Amplification of the large sub-unit (LSU) region of rRNA genes was performed using forward primer LR0R (5′-ACCCGCTGAACTTAAGC-3′) and reverse primer LR3 (5′-GGTCCGTGTTTCAAGAC-3′); PCR conditions consisted of an initial denaturation step at 94 °C for 5 min, followed by 30 cycles at 94 °C for 30 s, 55 °C for 30 s, 72 °C for 30 s, and a final extension at 72 °C for 10 min.

### 2.3. Phylogenetic Analysis

Sanger sequencing of the amplified products was performed by Bio-Fab research s.r.l (Rome, Italy). The nucleotide sequences were analyzed by BLAST (Basic Local Alignment Search Tool) using the National Center of Biotechnology Information (NCBI) database (https://www.ncbi.nlm.nih.gov/; accessed on 20 February 2023). Newly generated ITS and LSU rDNA sequences were deposited at the NCBI database under the accession numbers from OQ319946 to OQ319967 and from OQ302484 to OQ302507, respectively.

To perform a multi-locus tree, the ITS and LSU sequences for each strain were previously combined in FASTA format, obtaining a single sequence of about 1250 nucleotides (ITS:1-537 and LSU:538-1246 nucleotide positions). The new combined dataset was aligned with the ClustalW software included in the MEGA 11 package [38], with 19 combined sequences, downloaded from the NCBI database of meristematic fungi isolated from cultural heritage stone-works (Appendix A). The resulting alignment was checked manually and corrected if necessary. The software MEGA 11 was used to construct phylogenetic trees using the Neighbour-Joining method (NJ) [39]. Sequences divergences among strains were quantified with the Kimura-2-parameter distance model [40]. For treatment of gaps, the “Complete Deletion” option was chosen, and bootstraps analysis (1000 replicates) was used to test the topology of the NJ method data.

### 2.4. Thermal Preferences, Salt Tolerance, and Acid Test

After identification, representative strains of the isolated RIF were selected for physiological analysis.

A growth test at different temperatures was carried out by inoculating small fragments of mycelia on MEA no. 2 (VWR Chemicals, Milan, Italy) plates, in triplicate, and incubating them for 28 days at 5 °C, 15 °C, 20 °C, 30 °C, and 37 °C. Three biological replicas were performed.

A growth test at different NaCl concentration was carried out by inoculating small fragments of mycelia on MEA no. 2 (VWR Chemicals, Milan, Italy) plates with increasing concentration of NaCl (5%, 10% and 15% [*w*/*v*]), in triplicate, and incubating them for 28 days at 20 °C (RT).

In both growth tests, the mycelium was measured at the beginning (t_0_) and at the end (t_4_) of incubation, in two perpendicular dimensions; the total growth (t_4_–t_0_) was calculated as the average ± standard deviation (SD) of the six measured values (three for each size) for each strain. The diameter of the colonies was measured with a ruler.

To highlight the potential chemical damage they could cause by growing on carbonate rocks, isolated fungi were also tested for acid production. This was assessed by inoculating the mycelium of each strain on CaCO_3_ agar medium (yeast extract 5 g; glucose 50 g; CaCO_3_ 5 g; agar 15 g in a volume of 1 L; [20]), incubating the plates at 20 °C, and reading the presence of a halo after 28 days of incubation. Positive results were indicated by a clear zone around the fungal colony after flooding with Lugol (Merk KGaA, Darmstadt, Germany) [20].

### 2.5. Sensitivity Test to Selected Biocides

Two essential oils (EOs) were selected as innovative green biocides: essential oils (EO) of *Thymus vulgaris* L. (EO-Thymus) and *Origanum vulgare* (EO-Origanum) (Erbamea, s.r.l., Italy); their chemical composition was reported by Santo et al. (2023) [36]. BiotinT (BT) (CTS s.r.l., Florence, Italy), containing 2-*n*-octyl-4-isothiazolin-3-one and didecyl-dimethyl-ammonium chloride, was used as a traditional biocide. Stock solutions were prepared by disolving EOs in DMSO and BiotinT in water (both 25% [*v*/*v*]).

The sensitivity of the meristematic strains to the biocides was assessed by incorporating the appropriate volume of each stock solution into the MEA medium at varying final concentrations: 0.0025%, 0.005%, 0.012%, and 0.025% [*v*/*v* of medium]. Each concentration for each strain was tested in triplicate. The test was carried out by inoculating small fragments of mycelia on the medium and incubating them for 28 days (4 weeks) at 20 °C (RT). The mycelium was measured with a ruler at the beginning (t_0_) and at the end (t_4_) of the incubation in two perpendicular dimensions; the total growth (t_4_–t_0_) was calculated as the average ± standard deviation (SD) of the six measured values.

### 2.6. Statistical Analyses

The collected data about growth test at different temperatures and sensitivity to biocides were analyzed with a Two-way ANOVA (Analysis of Variance) test with the GraphPad Prism Version 9.2.0 software (GraphPad.com, accessed on 21 May 2023), and the Tukey-Kramer comparison was used to highlight statistically significant differences.

## 3. Results

### 3.1. Cultivation and Identification of Meristematic Fungi

The viable titer of meristematic fungi in the NW area was 1.2 × 10^6^ ± 2.41 × 10^5^ CFU/g of marble, while, in the SE area, it was found to be 5.30 × 10^5^ ± 6.08 × 10^4^ CFU/g after an incubation period of eight weeks. A total of 24 distinct morphotypes were identified, based on colony morphology, with 12 originating from the NW area and 12 from the SE area. DNA was extracted, and the ribosomal ITS and LSU phylogenetic markers were amplified and sequenced for each strain. The results of sequence analysis are reported in Table 1. For all the sequences, except for the ITS sequence of strains M7, M25, M32, and M38, an identity > 99% was found with known rDNA sequences classified at the level of genus or species (Table 1). For sequences of strains NT-S-5 and TW-N-Tq, no identity with known sequences at a deeper taxonomic level than class (Dothideomycetes), was found.

All the identified strains belonged to the classes of Eurotiomycetes (genera *Knufia* and *Lithohypha*) and Dothideomycetes (genera *Coniosporium*, *Neodevriesia,* and *Vermiconia*, strains NT-S-5 and TW-N-Tq).

### 3.2. Multi-Locus Phylogeny

Molecular phylogeny was performed to better evaluate biodiversity and to define the phylogenetic position of the strains with low or ambiguous identities. The two-gene dataset includes 43 strains, belonging to the Eurotiomycetes and the Dothideomycetes classes (Appendix A). Figure 2 shows the multi-locus phylogenetic tree generated. The tree was rooted with *Saccharomyces cerevisiae* MA09-AN (family Saccharomycetaceae), chosen as an outgroup.

The topology of the tree is in accordance with the most recent phylogenetic analyses on meristematic fungi [27,28].

Whitin the family Trichomeriaceae, class Eurothiomycetes, two groups were discernable. The first comprises the strains of the genus *Knufia.* Five strains were isolated from the marble of Florence Cathedral (M37, M43, m8, m16, and m17) cluster, with *Knufia marmoricola* strains having been isolated from marbles of Cortile della Pigna (Vatican City) and Santa Maria Cathedral of Cagliari (Italy). A further strain, m15, clustered with *Knufia petricola* strains, was isolated from a from a marble statue in Messina (Italy) and from a calcarenite obelisque in Corfù (Greece). In the basal position of the family Trichomeriaceae, seven strains (M1, M34, M35, M36, M41, m13, and m21) clustered with *Lithohypha guttulata* strains isolated from the marbles of Cortile della Pigna (Vatican City). The strain m20 clusters in a separate position, but it originated from the same node, which defines *Knufia* sp. and *Lithohypha* sp., supported with 92% bootstrap. Within the class Dothideomycetes, three families were at least discernable (incertae sedis, Paradevriesiaceae, and Extremaceae). Three strains of the *Coniosporium* genus (M7, M30, and M38) were assigned to *Coniosporium uncinatum* species because they cluster with *Coniosporium uncinatum* CBS 100219, supported by a 100% bootstrap. In the second group, two strains, NT-S-5 and TW-N-Tq, cluster together. The above-mentioned strains belonged to Dothideomycetes incertae sedis family (taxonomy not yet defined). M32 clusters with a *Paradevriesia compacta* strain (Paradevriesiaceae family) were isolated from a limestone in Mallorca (Spain). At least four strains (M25, M40, m10, and m11) cluster with *Vermiconia calcicola* strains retrieved from the literature, and they belong to the Extremaceae family.

### 3.3. Physiological Analyses on RIF Strains

Based on the phylogenetic analysis, eight strains, at least one for each genus, were chosen for physiological analyses: three strains of *Knufia,* showing three macroscopically different morphologies (m8, m15, and m16), a strain of *Lithohypha* (m21), one of *Coniosporium* (M30), and one of *Vermiconia* (m10). The phylogenetically ambiguous strain m20 and a representative of “unknown Dothideomycetes” (TW-N-Tq) were also included (Figure 3). The colony morphology of each selected strain is described in Appendix A. From Figure 3 and Appendix A, some morphological and growth characteristics of the different genera and different strains of the same genus (i.e., *Knufia*) are appreciable.

Regarding these strains, thermal preference, salt tolerance, and carbonate solubilization were tested. Results are expressed as single average size values, even if the measurement of the mycelium was taken in the two perpendicular dimensions of the colony surface, as described in Section 2.4. This choice was validated by the fact that the length and width values of colonies were proportional (Appendix A).

#### 3.3.1. Thermal Preferences

The strains showed different growth, according to the temperature (Table 2). All tested strains were able to grow between 5 °C and 30 °C. No strains grew at 37 °C. The best growth temperature was 20 °C for six strains (m10, m15, m16, m20, m21, and M30), 15 °C for m8, and 30 °C for TW-N-Tq.

#### 3.3.2. Salt Tolerance

All strains were not able to grow at concentrations greater than 5% NaCl (*w*/*v*) (Table 3). The average values obtained were compared with the growth value of the control (growth on standard MEA without salt addition). In all cases, growth was lower when compared with the control, so the percentage of decrease was calculated.

The most resistant strain to 5% NaCl concentration was m10, *Vermiconia* sp., while the least resistant, showing a complete inhibition of growth, were m21 (*Lithohypha*) and TW-N-Tq (Dothideomycetes).

To evaluate the resilience of the strains to resume growth after stressed conditions, after the incubation at 10% and 15% [*w*/*v*] NaCl for 28 days, the mycelia were transferred to MEA plates without salt. Strains m16 (*Knufia*) and m21 (*Lithohypha*) were unable to resume growth in both cases, while m8 (*Knufia*) was unable to recover its vital functions when transferred from medium with 15% NaCl. This, together with the high percentage of growth inhibition at 5% NaCl, showed by m8, m15, and m16 (Table 3), suggests that the genus *Knufia* is particularly sensitive to salinity. All the other strains showed growth when transferred on MEA salt-free medium.

#### 3.3.3. Carbonate Solubilization Test

Acid production was shown in seven out of eight tested strains by observing the presence of halos in the CaCO_3_ agar medium. No production was recorded for one of the three *Knufia* species (m15). Appendix A summarizes the results of the acid test and shows the images after treatment with Lugol’s solution for strains m16, m21, and m8.

#### 3.3.4. Sensitivity to Biocides

The effect of the EOs and BiotinT on the growth of meristematic fungi was tested. We started from 0.025% (*v*/*v*) as the maximum concentration, since it inhibited the growth of the whole cultivable community of bacteria and filamentous fungi, inhabiting the SMFC marble in in vitro sensitivity tests [36].

To evaluate the effect of the tested biocides on each meristematic strain, the average growth obtained on plates with biocides were compared with the control (growth on MEA with DMSO) (Table 4 and Appendix A).

EO-Origanum was effective in inhibiting the growth of meristematic fungi: at 0.025% no strain grew, and, at 0.012%, only three strains (m10, m20, and m21) out of eight were resistant. EO-Thymus inhibited the growth of five strains at 0.025% (m8, m15, m16, M30, and TW-N-Tq), with a high percentage decrease in growth for the remaining three strains (m10, m20, and m21). All the strains showed a lower sensitivity to BiotinT, since they could grow even at the highest concentration tested (0.025%). Effectiveness of EO-Origanum, EO-Thymus, and BiotinT on all the fungi tested is showed in Figure 4. At the highest concentration (0.025%), the EO-Origanum and EO-Thymus showed greater effectiveness when compared with BiotinT. At the lowest concentrations of 0.0025% and 0.005%, EO-Origanum and BiotinT had comparable inhibitory effects.

## 4. Discussion

In this study, we provide a comprehensive description of the community of meristematic fungi that reside on the external white marble of Florence Cathedral.

These fungi, known as RIF, were isolated from two areas of the Cathedral with exposure to NW and SE, where their contribution to the darkening of marble was demonstrated in a previous work [2]. The metataxonomic analysis revealed that RIF are the main component of the fungal community in these darkened areas [31]. However, cultivation conditions previously adopted to study the SMFC cultivable community were biased towards fast-growing filamentous fungi [2]. To address this, we optimized the cultivation conditions to better investigate the RIF community.

The characterization of RIF isolates was performed by analyzing their phylogenetic relationships and physiological characteristics. Additionally, a comprehensive evaluation of their susceptibility to various biocides, encompassing both conventional and innovative types, was conducted. Molecular taxonomy is necessary for meristematic fungi because of their poor morphological differentiation and heterogenous phylogenesis [4,25]. On the other hand, the knowledge of the ecological and physiological characteristics of the isolated microorganisms is essential for understanding their contribution to biodeterioration [6,20,21,41,42].

The viable titer of meristematic fungi in SMFC marble was found to be in the range of 10^5^–10^6^ CFU per gram, which is two orders of magnitude higher than the viable titer observed for filamentous fungi (10^3^–10^4^ CFU per gram) [2]. Although the incubation time for meristematic and filamentous fungi was different (fifty-six days and seven days, respectively) due to their different growth rates, this finding suggests that there is a significant presence of RIF on SMFC marble. It is noteworthy that the viable titer of RIF at the NW site was twice as high as that at the SE site, consistent with previous findings for filamentous fungi [2] and bacteria [31]. This difference may be attributed to the greater epilithic growth of the darkened biofilm at the NW site, which is favored by the higher water content of the north-exposed marbles [2].

Based on colony morphology, we identified 24 morphotypes, 12 from NW and 12 from SE. The taxonomic assignment was supported by identifying ITS rDNA region sequences that shared a similarity of over 99% with existing entries in the database [20]; the use of more markers, however, allows for a better phylogenetic definition. For this reason, we merged ribosomal ITS and LSU markers, generating a multi-locus phylogenetic tree with the 24 RIF isolated in this study, which were clustered with the other 19 RIF isolated from cultural stone artifacts (Figure 2).

The sequence analysis of the ITS and LSU markers revealed the presence of five genera and two classes based on the latest taxonomic classification [27,28]. We found genera *Knufia*, *Lithohypha*, *Coniosporium* (incertae sedis), *Paradevriesia*, and *Vermiconia*. They were evenly distributed within the SE and NW areas, with the exception of *Paradevriesia* (strain M32), which was only identified at the NW site. This indicates a wide diversity of RIF in the limited sampled marble areas, beyond what was revealed by the viable titer. It is interesting to note that *Knufia*, *Lithohypha* (formerly *Lithophila*), *Paradevriesia* (formerly *Devriesia*), and *Vermiconia* were identified among the most abundant fungal genera in SMFC marble using metataxonomic analysis [31].

At the level of species diversity, the two classes Eurotiomycetes and Dothideomycetes are similarly represented on SMFC marble. That would seem to contradict the tendency that had emerged by other studies of the predominance of the class Eurotiomycetes on heritage stones in urban open environments [21,23,43] and of the Dothideomycets in natural, unpolluted, and harsh environments [23,43]. For strains TW-N-Tq and NT-S-5, an identification at a deeper taxonomic level than class (Dothideomycetes) was not possible, since no similarities with rDNA sequences, classified at the level of genus or species, were found in NCBI/EMBO databases.

The topology of the tree is in accordance with the most recent phylogenetic studies [27,43], and it was useful to confirm or reconsider the isolated strains’ taxonomic assignment. The Dothideomycetes strains TW-N-Tq and NT-S-5, isolated at the NW and at the SE regions, respectively, cluster together, showing relationship with the Paradevriesaceae. They would be novel, similar fungi, whose characterization will be part of the ongoing work. The strain m15 clusters with strains of *Knufia petricola* isolated from stone, despite the BlastN best match with *K. marmoricola*, while the other *Knufia* strains (M37, m17, M43, m8, and m16) cluster with *K. marmoricola*. Moreover, the taxonomic classification of strain m20 (*Lithohypha*/*Coniosporium*, Table 1) remains ambiguous, also, in the phylogenetic tree, where it seems to belong to a new family. The phylogenetic analysis did not evidence a clear division between strains isolated from the NW and the SE, but it underlines the remarkable biodiversity at both sites.

Physiological tests were performed to differentiate metabolic traits of RIF, relating to their tolerance to environmental stressors and their interaction with stone [21]. A better understanding of their characteristics is important to design both preventive and corrective control measures to protect stone heritage [20,21].

The thermal preference test yielded information on the growth range and optimum temperature of the studied strains. All the tested RIF were able to grow within the range 5–30 °C, in agreement with what was found for *Knufia*, *Coniosporium,* and *Vermiconia* in the recent literature [21]. Overall, the tested strains showed a profile between cryophile and mesophile, with an optimal growth temperature between 15 °C and 20 °C. Even if no growth was observed at 37 °C after 28 days of incubation, such strains are able to cope with the thermal stress and the strong solar radiation, particularly on the SE areas of SMFC marble. Strategies adopted by rock-inhabiting black fungi under temperature and UV stresses include extreme slow microcolonial growth, heavily melanized cells, and a general downregulation of metabolic processes [15].

The salt test was proposed to demonstrate the salt tolerance of black meristematic fungi, as well as their general tolerance to extreme conditions [10,15,20,21]. We found that six strains out of eight were able to grow in the presence of 5% NaCl (*w*/*v*). Our results are in accordance with the findings obtained for the same genera by Isola et al. (2022) [21], with the difference that the *K. petricola* strain (CCFEE 5776) was able to grow up to 10% NaCl. It should be stressed, however, that, in the study by Isola et al. (2022) [21], RIF were isolated from monumental stones located close to the sea environment, with saltiness periodically deposited and washed away from surfaces with rains. Conversely, the majority of the SMFC genera tested exhibited a notable resilience, as they were able to resume growth after being exposed to >5% NaCl. This underscores their significant capacity to cope effectively with salt-induced stress.

The ability of meristematic fungi to lyse carbonate substrates through the production of organic acids had not emerged for years, and it was believed that the action of RIF on the stone was only of a mechanical nature [44]. However, this ability was observed in more recent studies [20,21]. Furthermore, in a study focused on *Knufia petricola* A95, an effective decrease in pH was observed around the colony [45]. In our study, seven out of eight strains were positive for the production of organic acids, showing a deteriorative potential that goes beyond the mechanical action they exert by growing on carbonate substrates [46]. *Lithohypha* strain m21 seems to have the greatest potential for this metabolic aspect, given the extent of the clear halo around the colony (Appendix A).

In recent years, the search for more sustainable strategies to control microbial growth has been developing in the field of cultural heritage conservation. Among natural products considered against biodeterioration of stone heritage, EOs have gained recognition as potential biocides due to their strong antimicrobial activity, and their experimental use has become increasingly popular as an alternative to conventional biocides [33,47]. The commonly used broad-spectrum commercial biocides can cause cross-resistance in microorganisms, and their use on stone materials may cause the drastic removal of competitors, favoring the development of more resistant microorganisms, also, in the surrounding environment [21,32]. Thus, traditional biocides are often used in sub-inhibitory concentrations, which, in the long run, could cause a decreasing sensitivity of the microorganisms to biocides and lead the polyextreme tolerant fungi towards an even greater resistance [48]. On the contrary, EOs appear to be relatively better in delaying microbial resistance mechanisms, also due to their complex and variable chemical nature (mainly aromatic compounds) [49,50,51]. Among the few studies on EO applications on stone, oregano and thyme EOs have been the most tested against microorganisms isolated from stone and against biofilm growing on stone surfaces (see references in Santo et al. (2023) [36]) due to their previous demonstrated efficacy in vitro.

In a previous work, we applied EO-Origanum and EO-Thymus on selected areas of the external white marble of Florence Cathedral. The results revealed that their effectiveness against the microbial community inhabiting the darkened marble was remarkably similar to that of the broad-spectrum commercial biocide BiotinT [36]. We performed preliminary tests on the SMFC cultivable microbiota to verify the effectiveness of the EO treatment, testing the sensitivity of the whole cultivable community of bacteria and filamentous fungi inhabiting the SMFC white marble [36]. EO-Origanum and EO-Thymus revealed to be very effective in the sensitivity test against bacterial and fungal microbiota cultivated from marble: no colonies grew at a concentration of 0.025% *v*/*v*, 80-fold lower than that of the working solutions used for marble treatment in situ. Traditional plate assays can be useful to study the sensitivity of metabolically active RIF colonies to treatments [21]. This aspect of RIF physiology has been, however, little explored in the literature, with only a few studies having been performed on conventional biocides [20,21]. In the present work, we assayed, for the first time, the sensitivity of RIF to thyme and oregano EOs by a plate test. The two EOs and the traditional biocide BiotinT were tested against RIF strains isolated from SMFC marble in a range of very low concentrations (0.0025–0.025%) with respect to those used for stone treatment. In this range, the tolerance to biocides was strain-specific and concentration-dependent. Considering the RIF response as a whole, EOs were more effective than BiotinT at the maximum concentrations tested, with EO-Origanum completely inhibiting all the tested strains at 0.025%. While the active principles of BiotinT are known, EOs are complex mixtures of substances with high chemical variability [49,50]. However, some antimicrobial effects can be imputable to single components of EOs, particularly terpenes [50]. The chemical analysis of the two EOs used in previous and in the present work showed that *T. vulgaris* EO was marked by the presence of high amounts of p-cymene and thymol, while carvacrol was very abundant in *O. vulgare* EO [36].

Our results add new insight to the tolerance to conventional and natural biocides of meristematic fungi, a group of microorganisms considered highly resistant to many types of chemical treatments [20,21]. Based on our findings, EO-Origanum and EO-Thymus demonstrate potential as sustainable and effective biocides for treating biological patinas caused by meristematic fungi. These results contribute to the development of eco-friendly and innovative solutions for the preservation of cultural heritage.

## 5. Conclusions

Black meristematic fungi have been identified as responsible for the diffuse darkening on the white marbles of the Cathedral of Santa Maria del Fiore in Florence (Italy) at two study sites. In the present work, the phylogenetic analysis of 24 isolated morphotypes, based on the sequences of the ITS and LSU rDNA regions, showed a wide diversity of the RIF in the sampled areas. The isolates belong at least to five different genera (*Knufia*, *Lithohypha*, *Coniosporium*, *Paradevriesia*, and *Vermiconia*). Two strains could not be identified at a taxonomical level lower than class, indicating monumental marble as an environment still little explored as source of novel strains, as has already emerged by the metataxonomic analysis of the SMFC community [31].

This research also gives an analysis of the main RIF traits with respect to thermal and salt growth preferences and on their possible deteriorating role while growing on carbonatic stones. Moreover, our study provides novel insights into the sensitivity of meristematic fungi to biocides, utilizing both essential oils and a commercial product.

## Figures and Tables

**Figure 1 jof-09-00665-f001:**
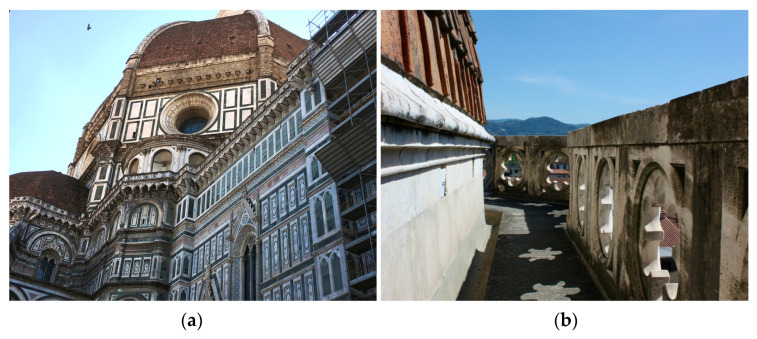
The Cathedral of Santa Maria del Fiore and the study sites. (**a**) The north-west side of the Cathedral with the external gallery running aroud the apses. (**b**) The inside of the openwork parapet of the external gallery at the south-east side, showing a widespread dark patina on marble.

**Figure 2 jof-09-00665-f002:**
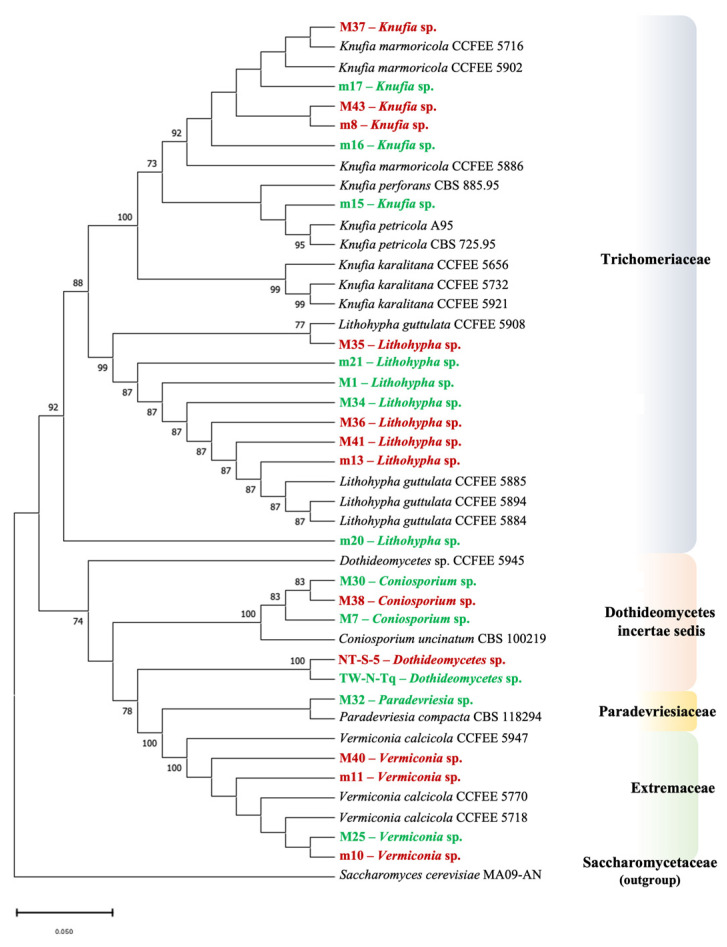
Two-gene multilocus phylogenetic tree constructed from the combination of ITS and LSU ribosomal sequences of 44 strains realized with Neighbour-Joining methods. Numbers indicate bootstrap confidence percentages. Bootstrap values above 70%, calculated from 1000 resampled data sets, are shown. The strains isolated from Florence Cathedral are indicated in bold, in red are identified the strains isolated from SE, and in green are identified the strains isolated from NW. The scale bar indicate the substitution per nucleotide. The accession numbers of sequences retrieved from the database are reported in Appendix A; the accession number of the *Saccharomyces cerevisiae* sequence (outgroup) is GQ458028.

**Figure 3 jof-09-00665-f003:**
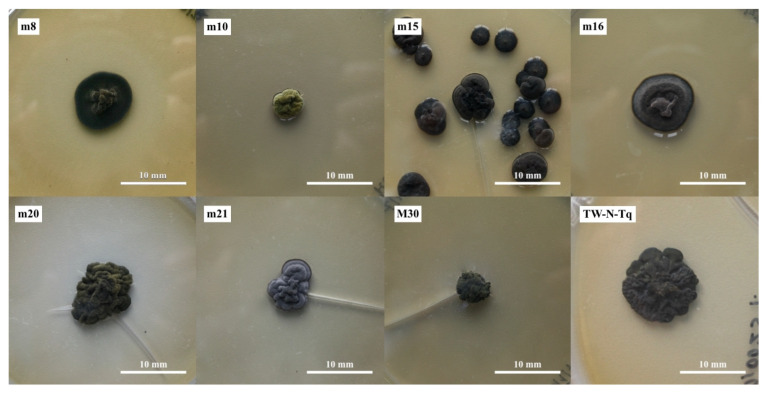
Colony appearance of the eight strains selected for physiological analysis on MEA medium. m8, m15, and m16, *Knufia* sp.; m10, *Vermiconia* sp.; m20, *Lithohypha*/*Coniosporium* ambiguous; m21, *Lithohypha* sp.; M30, *Coniosporium* sp.; TW-(N)-Tq, *Dothideomycetes* sp.

**Figure 4 jof-09-00665-f004:**
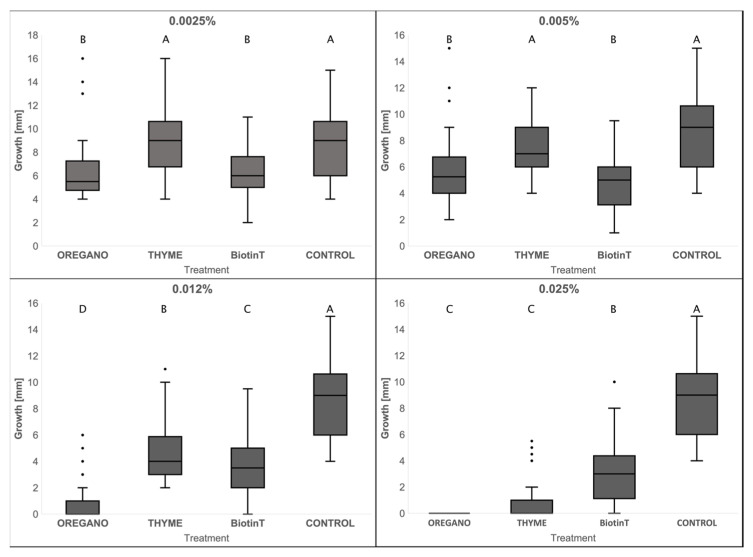
Average colony sizes (mm) of the eight selected meristematic strains for each treatment with biocides. Capital letters above each dot plot represent the Tukey’s test results: the same letters correspond to comparable averages of treatments, different letters indicate averages of treatments significantly different from each other. The control (DMSO) is always represented by the letter A, which necessarily indicates the treatment with the least inhibitory effect; the subsequent letters result in an increasingly significant inhibitory activity.

**Table 1 jof-09-00665-t001:** Identification of the meristematic strains isolated from Florence Cathedral marble by analysis of the ITS and LSU ribosomal regions. Each strain corresponds to a different morphotype.

Sampling Area	Strain	Organism with the Best ITS Match (Accession Number)	Identity (%)	Organism with the Best LSU Match (Accession Number)	Identity (%)
NW	TW-N-Tq	*Dothideomycetes* sp. (OQ319946)	99.45	*Dothideomycetes* sp. (OQ302485)	96.12
	M1	*Lithohypha guttulata* (MW361305)	100	*Lithohypha aloicola* (OQ302490)	100
	M7	*Coniosporium uncinatum* (OQ319953)	98.33	*Coniosporium uncinatum* (OQ302491)	100
	M25	*Vermiconia calcicola* (OQ319958)	98.20	*Vermiconia calcicola* (OQ302495)	100
	M30	*Coniosporium uncinatum* (OQ319954)	99.24	*Coniosporium uncinatum* (OQ302493)	100
	M32	*Paradevriesia pseudoamericana* (OQ319966)	95.94	*Paradevriesia americana* (OQ302498)	99.27
	M34	*Lithohypha guttulata* (OQ319948)	99.46	*Lithohypha aloicola* (OQ302486)	100
	m15	*Knufia marmoricola* (OQ319960)	99.58	*Knufia marmoricola* (OQ302499)	100
	m16	*Knufia marmoricola* (OQ319964)	100	*Knufia marmoricola* (OQ302500)	99.65
	m17	*Knufia marmoricola* (OQ319961)	100	*Knufia marmoricola* (OQ302501)	100
	m20	*Lithohypha guttulata* (OQ319952)	100	*Coniosporium uncinatum* (OQ302505)	99.52
	m21	*Lithohypha guttulata* (OQ319947)	99.81	*Lithohypha guttulata* (OQ302504)	100
SE	NT-S-5 *	*Dothideomycetes* sp. (MW361320)	99.45	*Dothideomycetes* sp. (OQ302484)	99.66
	M35	*Lithohypha* sp. (OQ319956)	99.44	*Lithohypha guttulata* (OQ302487)	99.65
	M36	*Lithohypha guttulata* (OQ319949)	99.81	*Lithohypha aloicola* (OQ302488)	99.83
	M37	*Knufia marmoricola* (OQ319965)	99.15	*Knufia marmoricola* (OQ302497)	100
	M38	*Coniosporium uncinatum* (OQ319955)	98.52	*Coniosporium uncinatum* (OQ302492)	100
	M40	*Vermiconia calcicola* (OQ319957)	100	*Vermiconia calcicola* (OQ302494)	100
	M41	*Lithohypha guttulata* (OQ319950)	99.82	*Lithohypha guttulata* (OQ302489)	100
	M43	*Knufia petricola* (OQ319962)	100	*Knufia marmoricola* (OQ302496)	100
	m8	*Knufia marmoricola* (OQ319963)	99.60	*Knufia sp.* (OQ302502)	99.65
	m10	*Vermiconia calcicola* (OQ319967)	100	*Vermiconia calcicola* (OQ302506)	99.65
	m11	*Vermiconia calcicola* (OQ319959)	99.18	*Vermiconia calcicola* (OQ302507)	99.82
	m13	*Lithohypha guttulata* (OQ319951)	99.81	*Lithohypha guttulata* (OQ302503)	99.82

* The isolation and ITS sequence of strain NT-S-5 were already reported by Santo et al. (2021) [2].

**Table 2 jof-09-00665-t002:** Results of the thermal preferences test. Measures of colonies size are expressed in mm ± SD. Colors indicates the growth (mm) from the lowest to the highest for each strain: orange < yellow < light green < green.

		Growth after 28 Days (mm ± SD)			
Strain	Genus	5 °C	15 °C	20 °C	30 °C *
m8	*Knufia* sp.	7.83 ± 0.98	9.75 ± 1.78	7.25 ± 2.53	1.38 ± 0.48
m10	*Vermiconia* sp.	3.92 ± 0.66	3.92 ± 0.38	5.02 ± 1.54	4.38 ± 0.75
m15	*Knufia* sp.	7.17 ± 0.93	8.33 ± 0.52	10.42 ± 1.20	4.83 ± 1.59
m16	*Knufia* sp.	8.50 ± 2.05	8.00 ± 0.71	13.00 ± 1.52	4.88 ± 1.31
m20	*Lithohypha* sp.	5.38 ± 1.46	4.95 ± 1.25	5.67 ± 1.17	3.88 ± 0.63
m21	*Lithohypha* sp.	6.13 ± 0.43	6.33 ± 1.51	14.00 ± 1.63	5.50 ± 2.12
M30	*Coniosporium* sp.	4.08 ± 1.20	5.17 ± 1.47	7.08 ± 0.92	4.50 ± 1.08
TW-N-Tq	*Dothideomycetes* sp.	6.67 ± 1.94	8.58 ± 2.42	9.17 ± 1.69	9.50 ± 2.68

* No growth was observed at 37 °C.

**Table 3 jof-09-00665-t003:** Results of the salt tolerance test. Measures of colonies’ sizes are expressed in mm ± SD.

		Growth after 28 Days (mm ± SD)		
Strain	Genus	Control	5% NaCl *	% Decrease in Growth
m8	*Knufia* sp.	8.75 ± 2.83	1.92 ± 0.49	78.1
m10	*Vermiconia* sp.	5.30 ± 1.56	4.08 ± 1.02	22.9
m15	*Knufia* sp.	11.75 ± 1.50	1.92 ± 0.92	83.7
m16	*Knufia* sp.	9.25 ± 0.80	1.92 ± 0.66	79.3
m20	*Lithohypha* sp.	5.00 ± 1.00	2.17 ± 0.52	56.7
m21	*Lithohypha* sp.	9.25 ± 0.63	0.00 ± 0.00	100.0
M30	*Coniosporium* sp.	9.00 ± 1.56	1.63 ± 0.75	81.9
TW-N-Tq	*Dothideomycetes* sp.	8.50 ± 1.42	0.00	100.0

* No growth was observed at the NaCl concentration > 5% (*w*/*v*) when tested.

**Table 4 jof-09-00665-t004:** Results of the sensitivity test to biocides after 28 days of incubation. The percentage decrease in growth was calculated for each strain by comparing the growth (mm) obtained on the plates with essential oils (EOs) to the growth (mm) in the control plates. For example, if the treated sample shows no reduction compared to the control, the value is 0%, and vice versa. 100% is fully inhibited. Growth data are available in Appendix A.

	% Decrease in Growth											
	EO-Origanum (% *v*/*v*)				EO-Thymus (% *v*/*v*)				BiotinT (% *v*/*v*)			
**Strain**	0.0025%	0.005%	0.012%	0.025%	0.0025%	0.005%	0.012%	0.025%	0.0025%	0.005%	0.012%	0.025%
m8	42	45	100	100	33	8	59	100	69	68	81	77
m10	0	0	77	100	0	0	0	79	0	0	12	26
m15	51	55	100	100	37	32	64	100	32	79	77	94
m16	25	28	100	100	0	12	44	100	28	40	72	72
m20	53	66	92	100	36	54	58	91	54	54	56	56
m21	0	0	56	100	0	4	2	57	6	18	31	35
M30	4	12	100	100	0	4	39	100	0	0	20	33
TW-N-Tq	38	43	100	100	20	10	57	100	37	45	75	98

## Data Availability

Not applicable.

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
