# Peer review of "Characterization of the Community of Black Meristematic Fungi Inhabiting the External White Marble of the Florence Cathedral"

_jof, 2023, doi:10.3390/jof9060665_

Round 1

Reviewer 1 Report

The manuscript entitled “Characterization of the community of black meristematic fungi inhabiting the external white marble of Florence Cathedral” deals with the study of the black meristematic fungi found at the Florence Cathedral. The authors isolated 24 strains from two exposed sites, which were identified and characterized based on the ITS and LSU regions. The authors also conducted physiological tests regarding temperature, salt tolerance and acid production for a few isolated strains. In addition, strains’ sensitivity to essential oils (thyme and oregano) and the biocide Biotin T was also tested.

Overall, this work is well structured and shows some interesting results, but it is confusing in some parts. I also suggest the authors to have their manuscript proofread by a native English speaker.

Bellow I suggest some improvements.

L33 – “(…) extreme conditions such as extreme temperatures (…). Please, replace one of the “extreme” words with another one.

L38 to L42 – “Even if RIF appear morphologically similar, they are phylogenetically heterogeneous, so, a phylogenetic analysis is often necessary to really estimate their biodiversity [4,16]. RIF are classified in the phylum of Ascomycota, in the two classes of Eurotiomycetes and Dothideomycetes. The most common orders are Capnodiales, Chaetothyriales and Mycosphaerellales [17–19]. Up to now more then 175 species have been identified [18].” I suggest the authors to transfer this paragraph to the end of line 55.

L42 – than, not then.

L45 – 20? 20 what?

L60 to L66 – The authors need to better contextualize these findings. Otherwise, it seems like the current study has already been conducted and it becomes very confusing. Every time I read this part and others in the Materials and Methods and Results, it feels like these results have already been presented on the works of Checcucci et al. 2022 and Santo et al. 2021.

L74 – tested, not testing.

L104 – “This sample was processed for cultivation of filamentous fungi and bacteria”. Again, this is one of the examples I was referring to previously and that is very confusing. The authors need to better contextualize all this.

L133 – The authors mention two forward primers. Where is the reverse primer?

L158 – “on criteria described in the Results section (3.3).”? Please delete this part.

L182 and L183 – “EOs were used as DMSO solution and BiotinT as water solution”. Do you mean that the solvents were DMSO and water?

L184 to 190 – the authors need to be clearer about the way they performed the sensitivity tests. How were the EOs and the Biotin T incorporated into the MEA medium? Were they spread onto the culture medium? Did you use discs? Please, provide more details on this.

L203 and L205 – “For all the sequences, except for strains NT-S-5 and TW-N-Tq, similarities with rDNA sequences classified at the level of genus or species showed an identity >99% in almost all cases (Table 1)”. Well, from my understanding of what is presented in Table 1, the strain NT-S-5 shows an identity of >99% for both ITS and LSU. Similarly, strain TW-N-Tq also shows an identity of more than 99% for the ITS region. On the other hand, strains M7, M25, M32 and M38 show identities bellow 99%. Please, clarify this and be more careful when presenting the results.

L218 – belonging.

L245 – 2-gene, not 2-genes.

L265 – “paragraphs 3.3.1, 3.3.2 and 3.3.4, respectively”. Please, delete this.

L271 to L281 – Please, re-write this paragraph, as it is very confusing in its current form.

L306 – out of.

L308/309 – Fig. S2 – what happened to the culture medium showed in a) for strain m8?

L323/324 – The authors need to explain in a better way the results presented on Figure 4. Also, I find the legend of Figure 4 to be very confusing.

L326 – L327 – “At the lowest concentrations of 0.0025% and 0.005% EO-Origanum and BiotinT had a comparable inhibitory effect, while the effect of OE-Thymus was comparable to that of the control.” This sentence is very confusing, please re-write this.

Table 4 – “Results regarding Biotin T and strain m21 showed a % of decreased growth of 352 at 0.025%.” Please, correct this.

L332 – What does the “(a)” at the beginning of the figure legend means??

L332 to L334 – “The percentage of decrease in growth was calculated for each strain comparing the results obtained on the plates with EOs with those of control plates (see Table S2).” This sentence is very confusing. Please, re-write this.

L348 – demonstrated.

L358 – essential to.

L411 – phylogenetic.

L419 – “(…) showed a different behavior towards temperature respect to the other tested Knufia strains (…). Please, re-write this.

L424 – in the presence.

L429 – able to grow.

L442 – microbial.

Discussion: Results regarding temperature, salt and acid production should be discussed in a deeper level. The same should be done for the rest of the discussion part, as in some cases it seems that the authors are only repeating the results they obtained (e.g., lines 471-486).

English language needs improvement. 

Reviewer 2 Report

This is an important and useful study of  black meristematic fungi inhabiting the external white marble of the Cathedral of Santa del Fiore in Florence.  This manuscript is a continuation of the research work of a team of authors devoted to the study of microorganisms that cause biodeterioration of the marble of the Cathedral in Florence. The topic presented by the authors seems interesting and useful.

I have the following comments which I hope will be helpful to the authors.

Section 2.1.

Lines 112-114. What morphological characteristics of the colonies were identified during the study of colonies in a stereomicroscope? How were morphotypes identified? Is it possible to give a brief description of these morphological features.

Section 3.1.

Lines 200-201. The same question about morphotypes. The colonies of black meristematic fungi are quite similar to each other on a macroscopic scale, how clearly the authors distinguished them and how these 24 morphotypes were identified. For example, strains m15, m16, m17 belonging to the same species of Knuphia marmoricola differed from each other in morphology. It would be good to clearly describe the morphology of the colonies.

Section 3.3. Figure 3.The size bar (10 mm) would be nice to include on all colony photos.

4. Discussion

Lines 373.  The same question about morphotypes. 24 morphotypes based on colony morphology were identified. But in the  manuscript there are no descriptions of these morphotypes, no descriptions of the morphology of the colonies.
